# COVID-19 Delta Wave Caused Early Overburden of Hospital Capacity in the Bulgarian Healthcare System in 2021

**DOI:** 10.3390/healthcare10040600

**Published:** 2022-03-22

**Authors:** Latchezar P. Tomov, Hristiana M. Batselova, Tsvetelina V. Velikova

**Affiliations:** 1Department of Informatics, New Bulgarian University, 1618 Sofia, Bulgaria; lptomov@nbu.bg; 2Department of Epidemiology and Hygiene, University Hospital “Saint George”, Medical University, 6000 Plovdiv, Bulgaria; dr_batselova@abv.bg; 3Department of Clinical Immunology, University Hospital Lozenetz, Sofia University St. Kliment Ohridski, 1407 Sofia, Bulgaria

**Keywords:** COVID-19, health system capacity, hospitalization risk, excess mortality

## Abstract

We develop and apply our methodology to estimate the overburdening of hospitals in Bulgaria during the upcoming delta surge. We base our estimations on an exponential risk model from the UK. Still, the methodology is generally applicable to all risk models, depending on age. Our hypothesis is that during the delta wave in Bulgaria, the system experienced a burden from late August due to decreased capacity. This will explain most of the excess mortality during the wave. We estimate the number of people from the active cases in need of hospitalization and intensive care.

## 1. Introduction

Coronavirus disease 2019 (COVID-19) continues to be a worldwide threat to mankind, requiring specialists, policymakers, and governments to address several issues that extend far beyond the health and well-being effects of this pandemic [1]. While the pandemic’s immediate health consequences play out, we need research and actions to be reconfigured to reduce risk, establish continuity, and enhance resilience for potential recovery [1].

We are witnessing conflicts in the ultimate social determinant of health as a result of the pandemic, which is producing a wide variety of challenges ranging from health policy constraints to worries about the distribution and access to healthcare [2]. Reorienting healthcare resources to COVID-19 management has reduced the capacity of already overburdened and underfunded health systems to handle other disease loads. Furthermore, discontinuing regular treatments and interventions and follow-up and immunization programs lead to outbreaks of avoidable transmissible illnesses, increased cancer incidence, and the number of challenging medical conditions in the late stages [1].

Furthermore, COVID-19 has grown difficult even for the most enduring healthcare systems. To stop the spread of the severe acute respiratory syndrome coronavirus 2 (SARS-CoV-2) coronavirus, extraordinary measures were implemented worldwide. Lockdowns appear to be one of the only successful interventions, although at the expense of economic slowdowns and restrictions on human liberties [2].

However, the current (fourth) wave in Bulgaria, typical of respiratory diseases, startled mankind once more and did more serious harm than the first ones. As a result, the optimism has started to fade. Importantly, developing herd immunity to control a pandemic through natural infection will undoubtedly result in millions of fatalities [3]. Furthermore, the delta variant of SARS-CoV-2 also raised people’s concerns, including healthcare workers [4]. 

At the outset of the pandemic, in this period of despair and lack of control over the illness, the discovery and implementation of COVID-19 vaccines provided us a glimpse of hope. It appears that one of the most effective public-health interventions—vaccination—may be used to tackle the pandemic [5].

COVID-19 challenged the entire health system in 2020–2021, especially emergency and hospital care. Emergency centers were on the frontline against COVID-19. As of 31 May 2021, 6511 employees work in centers for emergency medical care, of which 1129 are doctors, and 5382 are other staff. Compared to March 2020, the number of employees was reduced by 107 people [6]. The system of emergency medical care had good coverage on the territory of the country.

The total number of incoming calls to all emergency medical centers in the country for the period 1 March 2020–31 May 2021 is 936,517. Of these, 127,478 of them were COVID-19 disease-related calls, representing 13.61%, or each seventh call [6]. Tracked by months, the total number of entrants calls and those with COVID-19 correspond to the peaks in the total number infected in the country by months, the first wave of the pandemic (November–December 2020), and the second wave (March–April 2021). The largest number of calls was received in November 2020, March 2021, April 2021, and December 2020 (78,403; 77,188; 72,796; 70,164, respectively) [6].

The total number of days lost with temporary incapacity for work due to COVID-19 for all employees of the centers for emergency medical care for the period is 33,636 days, which is high and has a negative impact on the activity of the centers [6]. Furthermore, Klosiewicz et al. showed dynamics of the third wave of COVID-19 in a Polish single-center study. Therefore, new biomarkers are needed for decision making regarding medical treatment to shorten the duration of hospitalization in emergency departments, such as blood oxygen saturation [7]. 

According to the Bulgarian Hospital Association, the COVID-19 pandemic passed under the sign of a lack of a comprehensive strategy to deal with the health crisis. Now, a comprehensive strategy and a unified approach are also lacking in providing financial support and preparing for future pandemics [8].

The analysis of the emergency medical care system shows imbalances in insurance and usability. In addition, there are significant differences in the use of available teams’ financial resources and the volume of activity [8].

Real-time communication between medical professionals facilities and those of the centers for emergency medical care is carried out mainly by landline or simply not done due to lack of telephone line, which delays patient admission.

As part of the healthcare system in Bulgaria, hospital care was not prepared for the COVID-19 pandemic. Furthermore, the pandemic revealed weaknesses in the hospital care system, which have been known for years but have now reappeared. Hospital beds, their number, structure, and territorial distribution are indicators of security and access to hospital care.

Hospitalizations diagnosed with COVID-19 will begin in April 2020. However, the first cases in Bulgaria were confirmed in March [6]. This discrepancy is because since March, there is still no code to code the cases, and this code is being generated in the WHO in addition to the extension of the ICD–X revision [9].

The health authorities have tried to react according to the situation, especially partly for restructuring hospital structures. However, the time from March to September 2020 is not used well enough to take on more adequate measures and actions in view of the expected escalation of the pandemic in the autumn and winter months [6]. Moreover, in some districts, due to the initial low provision of the population with beds, almost half of the available hospital beds are being used to treat patients with COVID-19 [8].

While COVID-19 immunizations promise a restoration to usual healthcare standards, the vaccine-based approach requires enough population coverage and continuation of all measurements for limiting the pandemic. It necessitates appropriate policy, operational, and healthcare preparation. A problem is also the need for the readmission of COVID-19 survivors who rely on immunity against viruses due to natural infection. Moreover, it was shown that the wave of the multisystemic inflammatory syndrome in children is also anticipated after the factual “adult wave.”

Our hypothesis is that many people who need hospitalizations do not receive adequate and in-time hospital care due to changed admission rules. Therefore, the risk of hospitalization grows exponentially with age. We know that the share of people over 60 years is the main predictor for the ratio of hospitalized patients to active cases. The latter ratio should have at least a linear or slightly convex relationship with the first one. It should not tend to a constant value before the first ratio reaches 1 s. This is a basis for our methodology that allows us to qualitatively judge whether there is hospital overburdening and approximately estimate the need for hospital beds. Furthermore, our models for predicting deaths from cases by age groups show that mortality risk is also exponential, which reinforces the validity of the hospitalization risk model (deaths should be proportional to cases) and also shows a slight lag of 0 to 7 days between discovery and death for the 60+ age groups [10]. This small lag makes proper hospitalization in time crucial for controlling the mortality risk.

## 2. Methodology for Calculating the Number of Hospital Beds and ICU Beds Necessary in the Upcoming Surge

### 2.1. Estimating the Distribution of Hospitalizations per Age from New Cases per Age and an Exponential Risk for Hospitalization with Age 

If the distribution of new cases by age is f(t), where t is time in a year (age), we can obtain the distributions of the share of hospitalizations by age $f_{h}\left(t\right)$ by multiplying it with the function g(t): (1) gives the risk of hospitalization per age in percentage [11], and we need to normalize (2).
(1)$$g\left(t\right)=e^{0.044t}$$

This distribution varies very little with new cases day by day. This is due to the non-stationarity of the process and can be estimated in various stages of the pandemic, for example, for different dominating strains, by taking the cumulative cases for the period. The distribution for the total cumulative cases has a very stable value, because it reflects stable population demographics and contact patterns across ages that lead to this distribution of infected people, as a general picture of the hospitalizations. Since the data are not differentiated by sex, we expect to have bimodal distributions. We chose generalized Gaussian function (3) and estimated it with adjusted $R^{2}=0.984$, standard error (sum of squared residuals) SSE=0.0002386, and root mean squared deviation RMSE=0.008918.

The age groups in the data are 0–19, 20–29, 30–39, 40–49, 50–59, 60–69, 70–79, 80–89, and 90+. We take it from the Open Data Portal [10] and we evaluate by taking the middle of the intervals, with the points 10, 25, 35, 45, 55, 65, 75, 85, and 95 years of age (Figure 1).
(2)$$f_{h}\left(t\right)=f\left(t\right)\frac{g\left(t\right)}{100}$$
(3)$$f\left(x\right)=\;a_{1}e^{-{\left(\frac{x-b_{1}}{c_{1}}\right)}^{2}}+a_{2}e^{-{\left(\frac{x-b_{2}}{c_{2}}\right)}^{2}}=0.07207e^{-{\left(\frac{x-68.66}{13.96}\right)}^{2}}+0.1788e^{-{\left(\frac{x-\;46.57}{27.27}\right)}^{2}}$$

To treat (3) as probability density distribution, we need to scale with 10.4255, since for the appropriate age interval
(4)$$\int_{-\infty }^{\infty }f(t)dt=10.4255$$

To get our distribution, we need to integrate (3):(5)$$\int_{0}^{120}f_{h}\left(t\right)dt=13.15982285$$

This is the share of all individuals hospitalized up to 3 September 2021. So, for 46,0691 cases, this gives us 60,626.1 hospitalizations (Figure 2).

With this distribution, the share among 60+ is 66.66%, which is cumulative for 3.09%. The same share in the new cases is 52%. Thus, we can see disproportionally many patients over the age of 60 with the model of [11].

### 2.2. Using This Distribution in Simplified Form to Build a Model for the Beds and ICU Beds Depending on the Active Cases

#### 2.2.1. Using Single Number Instead of Distribution

The risk of hospitalization is exponential—*e*^0.044*t*^ [11], which allows us to use the population above a certain age threshold and its ratio to the whole population (in the new cases). Getting the mean risk from the integral of (1) divided by the length of the interval (0,100) gives us a mean risk of 18.2843% for the general population group. The mean risk for under 60 is 2.96%, which is under 1/6 of the total risk. For that reason, the ratio of people over 60% is enough to determine the hospitalization risk with good precision, to avoid using the whole distribution, which needs to be obtained from data for each day if we follow the approach, we described in Section 2.1.

#### 2.2.2. Modeling the Dependency between the Ratio of People over 60, the Active Cases, the Hospitalized Patients, and Patients in ICU

There will be a fixed share of hospitalizations for a specific fixed distribution of new cases by age. Instead of distribution, we use the ratio of people over 60 years. In this scenario, the model for the number of patients $H\left(a,r_{60+}\right)$, depending on active cases a and the ratio $r_{60+}$, is linear:(6)$$H\left(a,r_{60+}\right)=k_{60+}r_{60+}+k_{a_{c}}a_{c}+I$$

Such a linear model can be estimated with linear regression, given certain essential factors: The number of hospitalized patients is cointegrated with the number of active cases and the ratio of people over 60 years. This model can be estimated for various data ranges and would give different values (m). To fit this model, first, we need to prepare our data.

#### 2.2.3. Preparation of Data

We take active cases from the Open Data Portal [12].

The ratio of people over 60 we calculate for every new day. We differentiate the cumulative cases by age to obtain the daily cases by age, containing multiple periods. We analyzed frequency spectrums of active cases and this ratio to filter that noise and chose 21 days for the moving mean (Figure 3). We used 25 July2021 as the starting point of modeling when the ratio was lowest. The delta variant became dominant in Bulgaria (officially at the beginning of August, but data are delayed 7–14 days). This period of monotonic increase allows us to apply linear regression on cointegrated time series.

#### 2.2.4. Testing for Cointegration

We used the Engle–Granger cointegration test in Octave for the ratio $r_{60+}\left(t\right)$ and hospitalization cases *H*(*t*), and they are cointegrated with *p*-value = 0.001 < 0.05. For hospitalized and active cases (moving averages with the period of 21 days), the Johansen cointegration test [13] in Octave gives cointegration with a *p*-value = 0.001 < 0.05. Furthermore, we know in advance that hospitalizations are a certain percentage of new cases and total hospitalizations from active cases, depending mostly on that ratio from our analysis, which gives a theoretical background to our modeling efforts.

## 3. Results

### 3.1. Linear Regression with PCA—Hospitalizations

The correlation between the ratio $r_{60+}\left(t\right)$ and active cases a(t) is very high—95%, due to the stage of growing hospitalizations and growing ratio of 60+. Nonetheless, we need both variables, since we want to check how the coefficient $k_{h}$, representing the ratio between hospitalizations and active cases, changes with the increase of $r_{60+}\left(t\right)$. For that reason, we employ principal component analysis and make the regression with the transformed data. We first make a linear model for the whole interval to assess the fit. Then, we will use it by removing data points backward in time—from 3.09 15 points backward as a compromise between the number of observations needed for regression and the number of estimated coefficients.

The fitted model for hospitalizations with data from 25.07 to 3.09 has almost perfect characteristics (Table 1), as shown in Figure 4, including adjusted $R^{2}=1$. Here, $k_{60+}$ = 2583.2 and $k_{a}=0.13027$.

The model here says that on average, 13.027% of active cases become hospitalized above some number of patients, which depend linearly on the ratio $r_{60+}$. This is a linear approximation of a nonlinear relationship, which is similar to the approximation a tangent line makes to a curve around the point of tangency, or a plane, that is tangent to a surface—which is our case. We are constructing a plane in three dimensions—one is the active cases, the other is the ratio, and the third dimension is $H\left(a,r_{60+}\right)$ in (7). This is a local approximation of a surface. We can reconstruct the surface from a series of linear approximations—making this linear regression for m, m-1, m-2…m-14 points and obtaining the values for $k_{a_{c}}$, which represent the proportion of hospitalized patients and how it changes with $r_{60+}$.

#### Obtaining a Model for the Dependency between $r_{60+}$ and $k_{a_{c}}$

With the increase of $r_{60+}$, the coefficients $k_{a}$ increase toward a fixed value. The exponential nature of the risk with age suggests the lowest possible growth (as a limit) to be linear or the slowly aperiodic behavior of the process—the increase of the ratio of 60+ leads almost linearly to a value in an interval $\left[e^{0.044*60},e^{0.044*100}\right]=\left[14.01\%,81.45\%\right]$ with an average of 38.3169% (7a). A simple way to understand this is to imagine that the risks for people over and under 60 (7b) are uniformly distributed with age, and their expected risks fully represent them. Then, the combination of risks depends linearly on the ratio of people over 60. We get an approximation for the expected risk of the distribution of new cases (8). Thus, the choice of the general model is founded on knowledge from the risk model per age and is not just from the data. This is important, since the data contain risk and hospital capacity overburdening information. The theory allows us to distinguish them by justifying the selected method of extrapolation of data.

The general model is (10)—for hospitalizations and for ICU admissions alike, it is aperiodic, but its usage is as an almost linear model, with very low convexity (small exponential constant b)
(7a)$$M_{h}(age>60)=\frac{1}{100-60}\int_{60}^{100}e^{0.044t}dt=38.3169\%\;$$
(7b)$$M_{h}\left(age\leq 60\right)=\frac{1}{60-0}\overset{60}{\underset{0}{\int }}e^{0.044t}dt=4.92924\%\;$$
(8)$$M_{h}\approx M_{h}\left(age>60\right)*r_{60+}+\left(1-r_{60+}\right)M_{h}\left(age\leq 60\right)+O\left(n\right)$$
(9)$$k_{a_{c}}=k*\left(1-a*e^{-b*x+c}\right)$$

To calculate the results for hospitalization—we repeat the linear regression fit with decreasing time windows—3.09–25.07, 2.09–25.07….—15 consecutive times. Then, we fit model (8). For each of these points, we obtain good regressions with $R^{2}=1$ (Figure 5) and similar characteristics such as p-values and F statistics to those in Table 1. Then, we fit (9) with these data, and we obtain the model (10). It is applicable for $r_{60+}>0.05$:(10)$$fit_{mdl_{h\left(x\right)}}=k*\left(1-a*exp\left(-b*x+c\right)\right)=0.7945*\left(1-0.9197*exp\left(-0.811*x+0.1216\right)\right)$$

With that model, we can calculate for different numbers of active cases and different values of $r_{60+}$ the simple model (11), as shown in Figure 6.
(11)$$H\left(r_{60+},a_{c}\right)=k_{a_{c}}\left(r_{60+}\right)*a_{c}=k*\left(1-a_{c}*e^{-b*r_{60+}+c}\right)$$

### 3.2. ICU Beds

The exact same methodology is applied to ICU beds. We assume that the exponential risks of hospitalizations have a fixed proportion of cases needing intensive care and that this risk also grows exponentially. Thus, the general linear regression for the whole period is as good as it is for hospitalizations.

By using the same approach as for hospitalizations, we can fit (9) (Figure 7) and obtain (12) (Figure 8):(12)$$fit_{mdl_{icu\left(x\right)}}=k*\left(1-a*exp\left(-b*x+c\right)\right)=0.313\;*\left(1-0.933*exp\left(-0.3806*x+0.1341\right)\right)\;$$

### 3.3. Projections for Hospital Capacity Overburden, Based on That Methodology

Estimating the total number of people left out of the hospital system requires additional methodology. By comparing these models to the actual data, we can assess the difference between the number of expected hospitalizations and the number of actual hospitalizations. The same can be done for ICU patients. First, the overburdening of the system can be assessed visually from Figure 9 and Figure 10, which are the already fitted models with added data from 4.09 to 21.09. Second, the expected number of hospitalized patients and patients in the ICU can be calculated from the fitted models and the number of active cases (all data are with moving average with a period of 21 days) (Figure 11 and Figure 12).

## 4. Discussion

The present methodology has several limitations connected to the data. First and foremost, the exponential risk model we used is obtained from data in the UK, where the numbers of tests per 100,000 are several times higher. The system of contact tracing and isolation is much more thorough. The data, released in the open portal from the Ministry of Health, indicate that new hospitalizations follow the shape of the exponential model but are almost double, which is expected because Bulgaria has a double case fatality ratio of 4%, instead of the average 2%, which indicates that we find half of the infected in comparison with many other countries. However, our modeling is done so that we do not depend on that discrepancy between the model and actual new hospitalizations—we assess the ratio of the active cases and active hospitalizations, in which the errors in the numerator and denominator cancel out. Secondly, we assume that 21 days of averaging are enough to compensate for delays in corrections of active cases to avoid overestimating the hospitalizations ratio—if we have more official active cases than actual ones due to delays in correction.

The network dynamics model for the transmission of COVID-19, proposed by Zhu et al., showed a feasible approach to simulate the COVID-19 epidemic with different interventions, thus providing information and advice on how and when to open large-scale public facilities [14].

Next, we assume that the risk for ICU admission is also exponential and can be approached with the same linear model for the ratio of people over 60 years. Finally, we accept the discrepancies between model and actual cases as evidence for hospitals above capacity and not for evidence against the model.

Here, have some anecdotal evidence for limited capacity in hospitals in many regions of Bulgaria that were hit first in the delta wave, such as Targovishte and Burgas areas [15]. We also rely on the official policy of the Ministry of Health to transfer more cases into home treatment under supervision from general practitioners [16]. This is the core of our hypothesis—that the exponential risk of hospitalizations can be used to estimate the burden on hospitals and to provide evidence for exceeded capacity. Once again, it was confirmed that effective government communication is critical for public health crises [17], such as that observed in our country.

We show that there is a change after the alpha wave in the spring that leaves patients for home treatment and thus increases their mortality risk. The hospital system was left unprepared for the delta wave, leading to a lower percentage of people treated without delay. Instead of expanding the capacity for the expected large wave that came with two months delay, the system has shrunk it.

The lack of hospital beds and late hospitalization in some patients with COVID-19 leads to the late application of adequate treatment. This leads to an increase in severe disease cases, long COVID-19, and mortality. However, the problem with the lack of hospital beds is more due to the lack of trained medical staff to treat patients. This situation often leads to the need for urgent reorganization, which does not always comply with the rules for preventing nosocomial infections. As a result, the risk of healthcare-associated infection for both staff and patients increase. This creates fear in many patients to seek medical attention in a timely manner. These may be patients with COVID-19 or other diseases. This will inevitably complicate their condition and increase the risk of death. For this reason, increased mortality is expected in other diseases as well.

## 5. Conclusions

We propose a methodology to estimate how many patients from the active cases should be in hospital based on a model for the hospitalization risk. We use the exponential nature of the risk to simplify the task and use the ratio of people over 60 years old to all people in cases. We infer the deviation from the expected trajectory when both hospitalized patients and the ratio are growing and cointegrated with the active cases’ growth. We infer the age distribution of hospitalizations from that model and the age structure of cases.

Our work suggests that during the growth of delta cases in the summer, the policy of accepting patients changed and did not correspond to the exponential risk of hospitalizations, based on age. That implies the limited capacity of the hospital system and people left at home that should be accepted in the hospital. This is especially visible for intensive care beds when we have a decrease of occupied beds with the increase of the ratio of 60+ people in active cases. There is some uncertainty in estimating actual beds needed in our model. Still, its main goal is to show if there is a discrepancy between increases in active cases and hospitalized cases due to hospital system overburdening.

## Figures and Tables

**Figure 1 healthcare-10-00600-f001:**
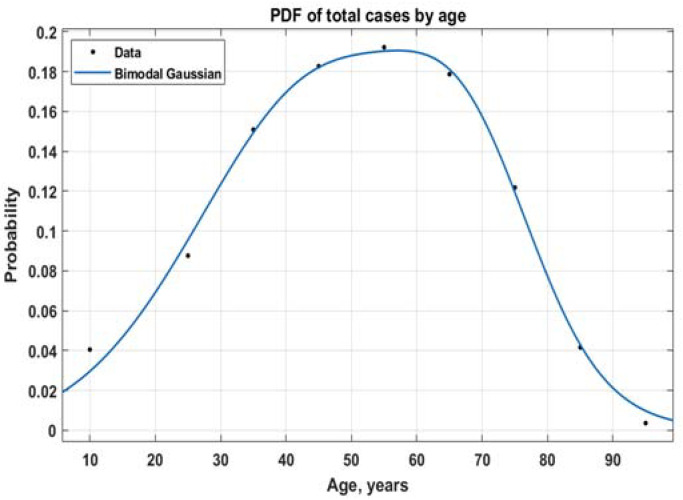
Probability density function of new cases by age for total cases up to 3 September 2021.

**Figure 2 healthcare-10-00600-f002:**
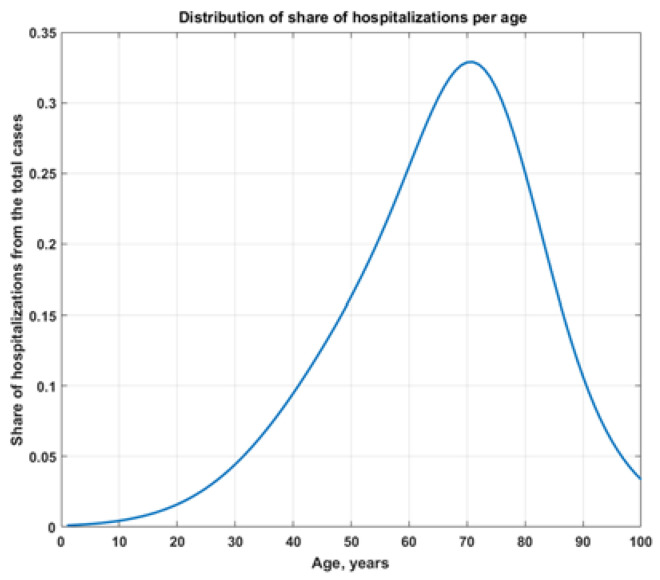
Distribution of the share of hospitalization from the total cases up to 3 September 2021 $f_{h}\left(t\right)$g.

**Figure 3 healthcare-10-00600-f003:**
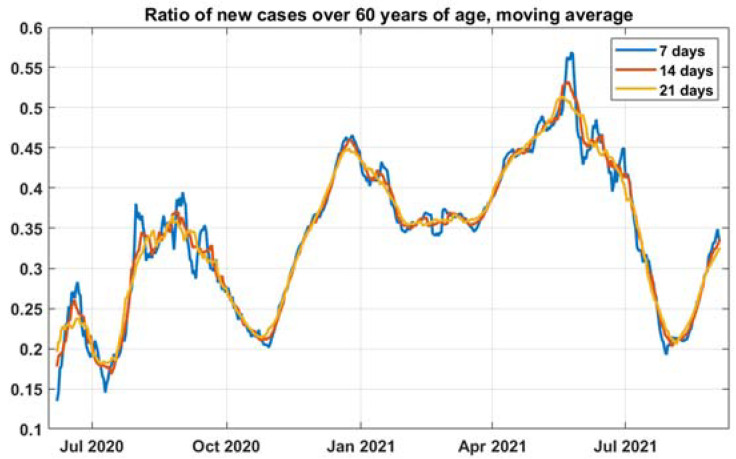
Comparison of 7, 14, and 21-days moving averages for *r*_60+_.

**Figure 4 healthcare-10-00600-f004:**
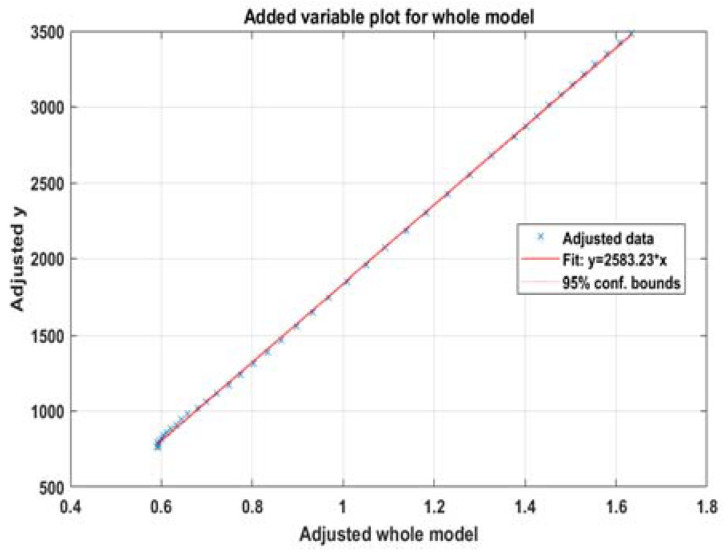
Model for (6) with hospitalizations for the whole period.

**Figure 5 healthcare-10-00600-f005:**
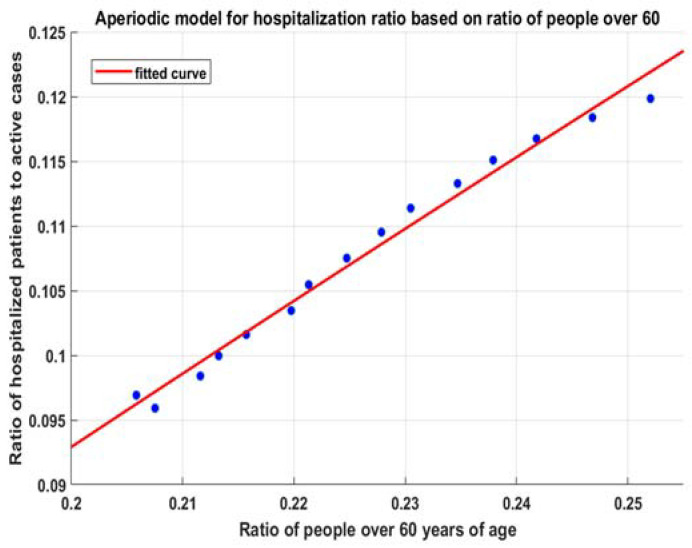
Fitted model (9) for hospitalizations.

**Figure 6 healthcare-10-00600-f006:**
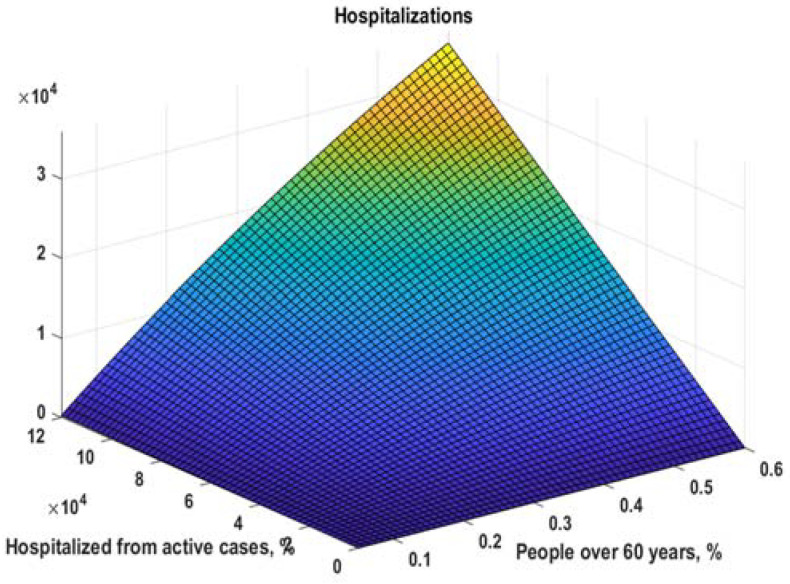
Model for the number of hospitalizations (11).

**Figure 7 healthcare-10-00600-f007:**
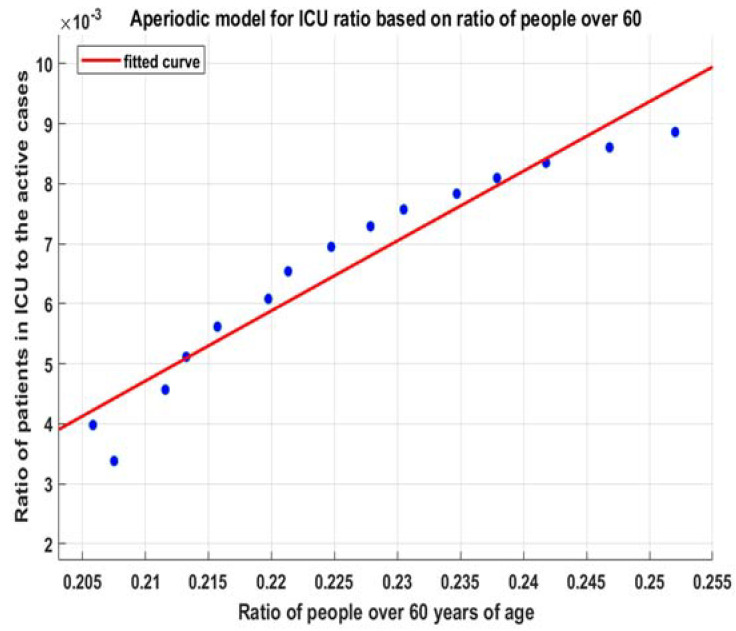
Fitted model (9) for hospitalizations.

**Figure 8 healthcare-10-00600-f008:**
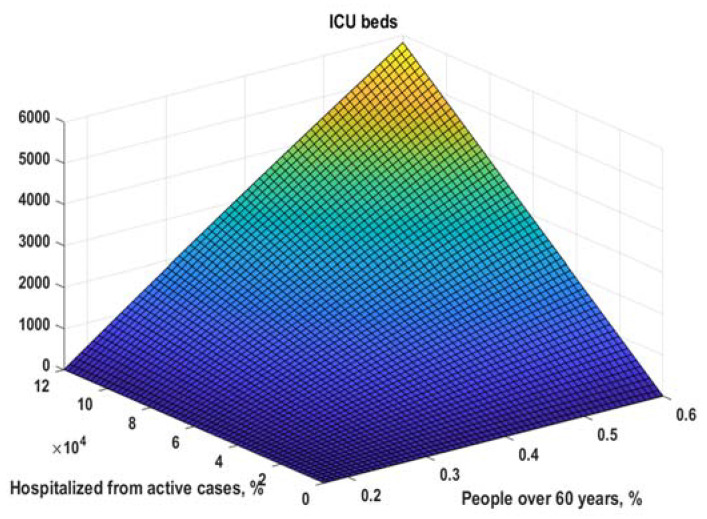
Model for the number of ICU beds (11). NOTE: Hospitalizations = all currently hospitalized patients, not daily new hospitalizations. The same is for ICU beds. All calculations here are for 21-day moving averages.

**Figure 9 healthcare-10-00600-f009:**
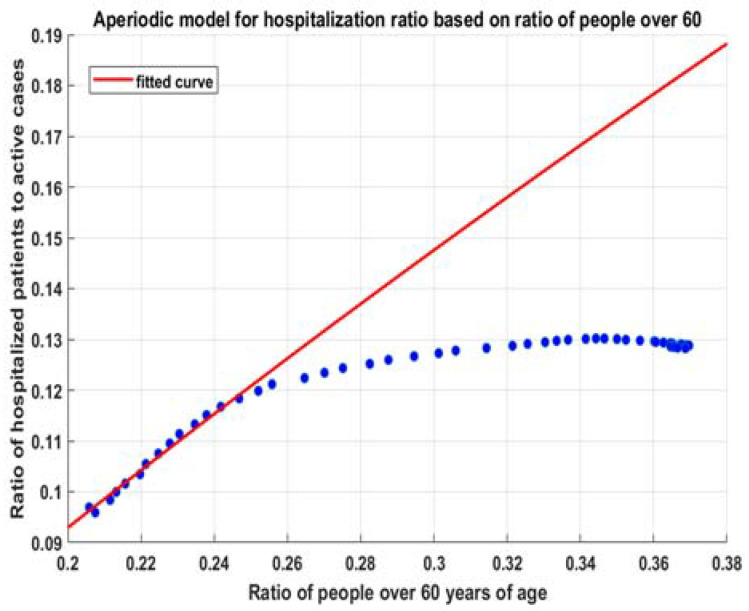
Deviation from the model for hospitalizations from 4 to 21 September 2021.

**Figure 10 healthcare-10-00600-f010:**
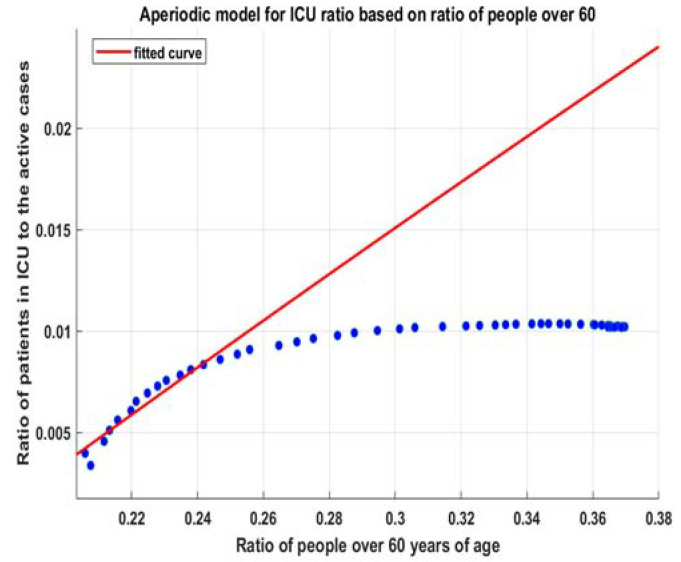
Deviation from a model for ICU in the period of 4 to 21 September 2021.

**Figure 11 healthcare-10-00600-f011:**
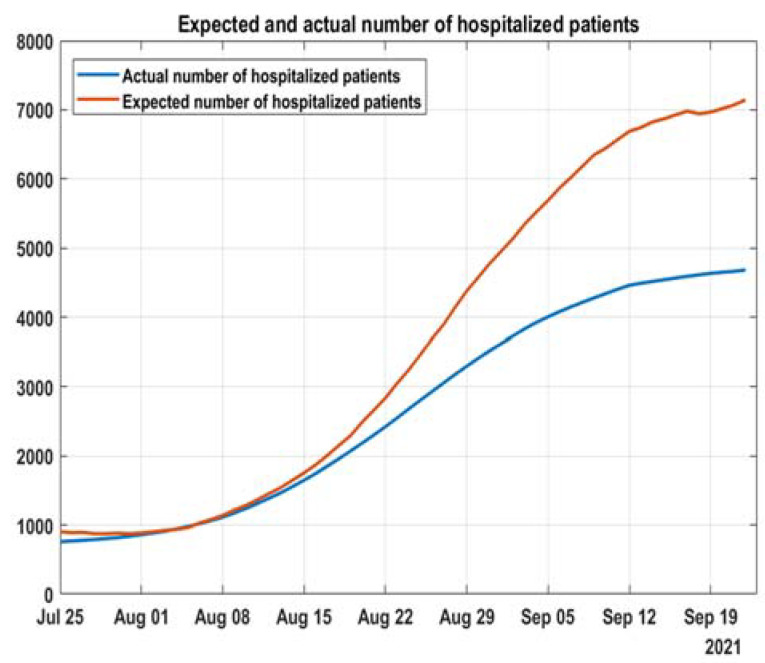
Discrepancy between model and actual hospitalizations.

**Figure 12 healthcare-10-00600-f012:**
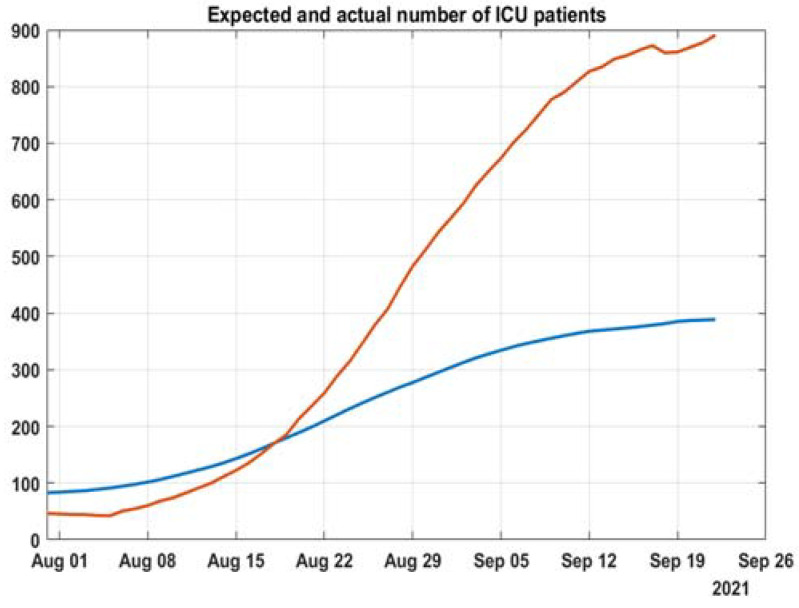
Discrepancy between model and actual ICU patients.

**Table 1 healthcare-10-00600-t001:** Results from linear regression with PCA for hospitalizations.

	Estimate	SE	tStat	*p* Value
(Intercept)	−744.64	34.793	−21.402	3.6842 × 10^−23^
×1	0.13027	0.0003206	406.33	2.4057 × 10^−72^
×2	2583.2	204.43	12.636	2.2938 × 10^−15^

## Data Availability

Data available on request due to restrictions, e.g., privacy or ethical. The data presented in this study are available on request from the corresponding author.

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
