# Peer review of "COVID-19 Delta Wave Caused Early Overburden of Hospital Capacity in the Bulgarian Healthcare System in 2021"

_healthcare, 2022, doi:10.3390/healthcare10040600_

Round 1

Reviewer 1 Report

Review of the paper: healthcare-1582699:

 Hypothesis: The delta wave caused early overburden of hospital capacity in the Bulgarian healthcare system

In this paper, the author (s) are interested in estimating the number of hospitalized patients in hospitals during a critical period of COVID-19 spreading (delta wave). For that aim, an exponential risk model is selected and mathematical results are proposed. The process of tuning the latter models is carried out by fitting with existing data. The obtained mathematical results are clearly explained. Several comparisons with the actual data are conducted and some limitations are mentioned. The overall scientific contribution is satisfactory. However, English language and editing errors need more corrections and improvements. A sample of such errors is presented below.

Based on the above comments, I recommend the acceptance of the paper provided that the English language and editing errors are corrected.

  1. Line 33: replace “… leads…” by “… lead…”.
  2. Line 50: replace “…challenged the intire health system …” by “…challenged the entire health system …”.
  3. Line 51: replace “Emergency centerswere in fronline against …” by “Emergency centers were in frontline against …”.
  4. Line 51: replace “…19.Until 31.05.2021….” by “…19. Until 31.05.2021….”.
  5. Line 52: replace “…6511 employees work inceneters for emergency ….” by “…6511 employees work in centers for emergency ….”.
  6. Line 63: replace “…(78,403, 77,188, 72,796, 70,164 respectively) ….” by “…78,403; 77,188; 72,796; 70,164 respectively ….”.
  7. Line 82: replace “…in the system hospital care, ….” by “…in the hospital care system,  ….”.
  8. Line 109: replace “…allows us to ….” by “…allow us to ….”.
  9. Line 114: replace “…crucial for the contro of ….” by “…crucial for the control of ….”.
  10. Line 114: replace “…(3) and estimated it with adjusted ….” by “…(3) and estimate it with adjusted ….”.
  11. Line 159: are you sure about “100?(?)g” ?.
  12. Line 172-173: rephrase “…if we 172 follow the described by us approach….”.
  13. Line 175: remove “and the”.`
  14. Line 185: replace “To fit this model that first we need..” by “To fit this model, first we need…”.
  15. Lines 280-281, 285: The number (11) for the presented equation is already used in line 272.

Author Response

Review of the paper: healthcare-1582699:

Hypothesis: The delta wave caused early overburden of hospital capacity in the Bulgarian healthcare system

In this paper, the author (s) are interested in estimating the number of hospitalized patients in hospitals during a critical period of COVID-19 spreading (delta wave). For that aim, an exponential risk model is selected and mathematical results are proposed. The process of tuning the latter models is carried out by fitting with existing data. The obtained mathematical results are clearly explained. Several comparisons with the actual data are conducted and some limitations are mentioned. The overall scientific contribution is satisfactory. However, English language and editing errors need more corrections and improvements. A sample of such errors is presented below.

Based on the above comments, I recommend the acceptance of the paper provided that the English language and editing errors are corrected.

  • Thank you for the overall evaluation of our paper as good. We addresses all the raised issues below in the revised version. Thank you for noticing them and giving us chance to revise our paper.
  1. Line 33: replace “… leads…” by “… lead…”.
  2. Line 50: replace “…challenged the intire health system …” by “…challenged the entire health system …”.
  3. Line 51: replace “Emergency centerswere in fronline against …” by “Emergency centers were in frontline against …”.
  4. Line 51: replace “…19.Until 31.05.2021….” by “…19. Until 31.05.2021….”.
  5. Line 52: replace “…6511 employees work inceneters for emergency ….” by “…6511 employees work in centers for emergency ….”.
  6. Line 63: replace “…(78,403, 77,188, 72,796, 70,164 respectively) ….” by “…78,403; 77,188; 72,796; 70,164 respectively ….”.
  7. Line 82: replace “…in the system hospital care, ….” by “…in the hospital care system,  ….”.
  8. Line 109: replace “…allows us to ….” by “…allow us to ….”.
  9. Line 114: replace “…crucial for the contro of ….” by “…crucial for the control of ….”.
  10. Line 114: replace “…(3) and estimated it with adjusted ….” by “…(3) and estimate it with adjusted ….”.
  11. Line 159: are you sure about “100?(?)g” ?.
  12. Line 172-173: rephrase “…if we 172 follow the described by us approach….”.
  13. Line 175: remove “and the”.`
  14. Line 185: replace “To fit this model that first we need..” by “To fit this model, first we need…”.
  15. Lines 280-281, 285: The number (11) for the presented equation is already used in line 272.
  • All issues were corrected. We encountered some other mistakes and tried to fix them all.

Reviewer 2 Report

MINOR REVISION

  1. The authors work highlights an important area in managing resource use in the covid-19 pandemic. The article contributes a lot to the knowledge in this important area. I have only minor suggestions. Would the authors consider taking out the word "Hypothesis" from the title? and phrase it differently?
  2. The paragraph on page 3, lines 103-115 detailing the hypothesis could be more concise and the hypothesis phrased more clearly.

Author Response

MINOR REVISION

  1. The authors work highlights an important area in managing resource use in the covid-19 pandemic. The article contributes a lot to the knowledge in this important area. I have only minor suggestions. Would the authors consider taking out the word "Hypothesis" from the title? and phrase it differently?
  2. Thank you very much for the valuable notes and for the overall evaluation of our paper as good. We accept the referee's suggestion to remove the word hypothesis from the title. We propose the following title:

”COVID-19 Delta wave caused early overburden of hospital capacity in the Bulgarian healthcare system in 2021”

  1. The paragraph on page 3, lines 103-115 detailing the hypothesis could be more concise and the hypothesis phrased more clearly.
  • Thank you for the suggestion. We revised the paragraph to make it more comprehensive and concise.